# Autonomous Planning of Discontinuous Terrain-Dependent Crawling for Space Dobby Robots

**DOI:** 10.3390/s23063334

**Published:** 2023-03-22

**Authors:** Jiabo Jiang, Cheng Wei, Yunfeng Yu, Shengxin Sun

**Affiliations:** 1School of Astronautics, Harbin Institute of Technology, Harbin 150001, China; 22b918071@stu.hit.edu.cn (J.J.); weicheng@hit.edu.cn (C.W.); 2Aerospace System Engineering Shanghai, Shanghai 200000, China

**Keywords:** space dobby robot, discontinuous terrain environment, autonomous planning, artificial potential field method

## Abstract

Complex space missions require more space robotic extravehicular operations required to crawl on spacecraft surfaces with discontinuous features at the graspable point, greatly increasing the difficulty of space robot motion manipulation. Therefore, this paper proposes an autonomous planning method for space dobby robots based on dynamic potential fields. This method can realize the autonomous crawling of space dobby robots in discontinuous environments while considering the task objectives and the self-collision problem of robotic arms when crawling. In this method, a hybrid event–time trigger with event triggering as the main trigger is proposed by combining the working characteristics of space dobby robots and improving the gait timing trigger; the dynamic potential field function is designed to adjust the space robot robotic arm grasping point adaptively according to the space robot state. Simulation results verify the effectiveness of the proposed autonomous planning method.

## 1. Introduction

With the increase in space exploration missions, space robots will gradually become the main bearers of future space exploration and space missions [1,2]. Traditional space robots are mainly single-armed and have poor stability and limited carrying capacity [3]. Therefore, space dobby robots are gradually becoming a research hotspot in the space robotics field [4,5].

Many scholars have studied space dobby robots. A novel path planning strategy, the distorted configuration space (DC-space) method, was proposed and proven by XIE to outperform the traditional search-based methods in terms of computational efficiency [6]. Yan proposed a multi-objective configuration optimization scheme for the dual-arm space robot in the pre-contact stage to maximize the operability and minimize the basic disturbance [7]. Xu proposed a method for two-arm space robots to grasp the optimal configuration of a rolling target based on task compatibility configuration to achieve the rapid stability of the rolling target [8]. Dang designs an obstacle avoidance method based on computer graphics which considers the robot body and end effector [9]. In order to solve the real-time problem of trajectory planning, Wu proposed a model-free reinforcement learning strategy for training the online trajectory planning strategy, so that the space robot can quickly schedule and execute actions [10]. Based on the position/force control strategy, Wang proposed a potential on-orbit screw-driven compliant operation strategy implemented by a dual-arm space robot [11].

However, the current research on space robots is mainly focused on the cooperative manipulation of multiple robot arms on the target object, whereas the research on space crawling, motion combining, task planning, and control is lessened.

The crawling motion of space dobby robots on dependent objects requires human involvement in the control closure when performing space manipulation tasks, which greatly limits the working effectiveness of space robots [12]. Therefore, the autonomous planning of the crawling motion of space dobby robots in discontinuous environments needs to be investigated.

Due to their similar structure, research on the crawling motion of space robots can refer to the research on legged robots. Payandeh investigated the effect of the longitudinal lateral motion of robots on the smooth motion of quadruped robots [13]. Wang Peng investigated the link between the motion stride and smoothness of quadruped robots [14]. Wang designed the static gait of quadruped robots based on the stability margin calculated by the center-of-pressure method and realized the stable walking of the quadruped robot in uneven terrain [15]. Lee refined the problem of obstacle avoidance of the quadruped robot and used the idea of the artificial potential field method to consider the ability of the quadruped robot to straddle obstacles for path planning [16]. Zhang analyzes the motion of a quadruped robot in rugged terrain and proposes a free gait generation method to achieve autonomous robot navigation with sufficient stability margins [17]. Xuesong investigated a motion planning algorithm for path planning, gait generation, gait conversion, and landing point search for terrain containing convex obstacles and exclusion zones to improve the adaptability of quadruped robots to complex environments [18]. Liang used the three-dimensional quasi-static equilibrium support region (3D QESR) as the constraint of the planning method for complex terrain and realized the quasi-static stable motion of the hexapod robot [19]. XuPeng treats the six-legged robot gait and foothold planning as a sequence optimization problem for the sparse drop point environment and uses the Monte Carlo tree search (MCTS) algorithm to optimize the entire traversal motion sequence to achieve balanced movement in a harsh environment [20].

However, space robot activities in space are mostly in discontinuous environments, and the above studies have not considered the motion in more demanding terrain environments (e.g., the outer surface of spacecraft with discontinuous graspable points), and it is important to study the autonomous planning of space robots in discontinuous terrain environments in space due to the effect of microgravity and the fact that the mechanical arm has much more DOFs than the single foot of a quadruped robot.

In this paper, the truss environment is used as the crawling environment of the space robot, and the autonomous planning of the dependent crawling in the discontinuous environment of the space robot is realized through the operations of gait trigger design, autonomous planning based on the dynamic potential field method, and whole-body controller design.

## 2. Space Dobby Robot Model

### 2.1. Space Dobby Robot Mechanism

The basic structure of a space dobby robot is divided into a substrate torso and a multi-branch chain robotic arm mechanism. For orbital maneuvering, attitude adjustment, and other whole-star movements, mostly actuators such as thruster systems and flywheel systems attached to the space robot substrate are used to drive. For maintenance, assembly, and other fine task movements, the robotic arm is needed to drive the robotic arm joints by hydraulic or motor drives, so that the robotic arm end load can complete fine operations.

Considering that the space robot has a wide variety of task requirements and needs more operating robotic arms to complete multiple task requirements while taking into account maneuverability and preventing itself from collision, a detachable structure is used as the connection between the robotic arm and the substrate. For the crawling motion requirements involved in this paper, a minimum of four robotic arms are required for execution. Considering other possible sub-tasks, the space robot needs to be equipped with six manipulators.

The mechanism diagram of the space dobby robot is shown in Figure 1. The main composition is composed of a substrate and six seven-DOF robotic arms. The substrate can be controlled by thrusters and flywheels for position and attitude control, and the robotic arms and corresponding hand claws are driven by the torque generated by motors.

Easy to describe, define the body coordinate system {B}; the origin is located in the center of mass of the robot body, the *Z*-axis direction points to the side of the matrix that deviates from the crawler, the *X*-axis direction points to the center of the end face that the forward direction passes through, and the *Y*-axis direction satisfies the right-hand rule. The global coordinate system {O} is defined to coincide with the initial state ontology coordinate system. The installation coordinate system {H} of the manipulator is defined. The origin is located on the robot body and connected to the manipulator. The direction of the coordinate axis is the same as that of the global coordinate system. The grasping coordinate system {P} of the gripper is defined. The origin is located at the center of the elliptical arc in the gripper. The *Z*-axis direction points to the grasping direction of the gripper, the *X*-axis direction points to the normal direction of the grasping surface, and the *Y*-axis direction satisfies the right-hand rule.

Due to the microgravity environment in space, the robot arm can be designed as an equally thick robot arm rod, and the end of the robot arm can be flexibly selected to carry tools according to the mission.

In the dynamics simulation software MBDyn, the virtual prototype model of the space dobby robot is established, and the model parameters are set according to the model in MBDyn to obtain the model shown in Figure 2. The relevant parameters are shown in the Table 1.

### 2.2. Dynamics Modeling

#### 2.2.1. Kinematic Model

In Cartesian space, the substrate of the space dobby robot can be regarded as a single rigid body with six DOFs, i.e., X-direction movement, Y-direction movement, Z-direction movement, rotation about the *X*-axis, rotation about the *Y*-axis, and rotation about the *Z*-axis. The single robot arm of the space dobby robot has seven DOFs; therefore, under the control of the seven-freedom robot arm, the hand claw of each robot arm has six DOF in Cartesian space. The position posture can be adjusted freely.

Referring to Wen-Hong [21], the formal expression of the velocity matrix of the end position of the robot arm is obtained.
(1)$$\left[\begin{array}{c} v_{e}\\ w_{e} \end{array}\right]=\left[\begin{array}{cc} I_{3} & {\hat{R}}_{eb}{}^{T}\\ 0 & I_{3} \end{array}\right]\left[\begin{array}{c} v_{b}\\ w_{b} \end{array}\right]+\left[\begin{array}{c} J_{Te}\\ J_{Re} \end{array}\right]\dot{q}$$
where $\left[\begin{array}{c} v_{e}\\ w_{e} \end{array}\right]$ is the end position speed of the robot arm, $\left[\begin{array}{c} v_{b}\\ w_{b} \end{array}\right]$ is the positional velocity of the space robot substrate, $J_{Te}$ represents the translational influence matrix of each joint of the robot arm, $J_{Re}$ represents the rotation influence matrix of each joint of the robot arm, and ${\hat{R}}_{ib}{}^{T}$ is the fork product matrix representing the vector of the origin of this system pointing to the end of the robot arm.

Further, the derived robot arm rod kinematic expression is
(2)$$\left[\begin{array}{c} v_{i}\\ w_{i} \end{array}\right]=\left[\begin{array}{cc} I_{3} & {\hat{R}}_{ib}{}^{T}\\ 0 & I_{3} \end{array}\right]\left[\begin{array}{c} v_{b}\\ w_{b} \end{array}\right]+\left[\begin{array}{c} J_{Ti}\\ J_{Ri} \end{array}\right]\dot{q}$$
where $\left[\begin{array}{c} v_{i}\\ w_{i} \end{array}\right]$ is the positional velocity of the linkage of robot arm No. *i*., $\left[\begin{array}{c} v_{b}\\ w_{b} \end{array}\right]$ is the positional velocity of the space robot substrate, $J_{Ti}$ represents the translational influence matrix of each bar of the robot arm, and $J_{Ri}$ represents the rotation influence matrix of each bar of the robot arm. Additionally, ${\hat{R}}_{ib}{}^{T}$ is the fork product matrix of the vector with the origin of this system pointing to the linkage of robot arm No. *i*.

#### 2.2.2. Kinetic Modeling Based on Lagrange Equations

The derivation of the kinetic model is conducted using the Lagrange method:(3)L=T−V
where L is the Lagrange function, T is the kinetic energy of the system, and V is the potential energy of the system. Considering that the operating environment is space and neglecting the effect of microgravity, the Lagrange expression can be simplified as follows:(4)L=T

Taking the derivative of time with respect to the Lagrange equation yields:(5)$$\frac{d}{dt}(\frac{\delta L}{\delta \dot{\varphi }})-\frac{\delta L}{\delta \varphi }=\frac{d}{dt}(\frac{\delta T}{\delta \dot{\varphi }})-\frac{\delta T}{\delta \varphi }=Q$$
where φ is the generalized coordinate and Q is the generalized force of the system.

The kinetic energy of the space robot system is expressed as
(6)$$T=T_{b}+\sum_{i=1}^{n}T_{i}=\frac{1}{2}\varphi^{T}H\left(\varphi \right)\varphi$$
where $T_{b}$ denotes the kinetic energy of the base of the space robot, $T_{i}$ denotes the kinetic energy of linkage *i* of the space robot arm, and H(φ) is the generalized mass matrix. The general kinetic equation can be obtained:(7)$$H\left(\varphi \right)\ddot{\varphi }+\dot{H}\left(\varphi \right)\dot{\varphi }-\frac{1}{2}{\dot{\varphi }}^{T}\frac{\delta H\left(\varphi \right)}{\delta \varphi }\dot{\varphi }=Q$$
In the above equation, let $C\left(\varphi ,\dot{\varphi }\right)\dot{\varphi }=\dot{H}\left(\varphi \right)\dot{\varphi }-\frac{1}{2}{\dot{\varphi }}^{T}\frac{\delta H\left(\varphi \right)}{\delta \varphi }\dot{\varphi }$.
(8)$$H\left(\varphi \right)\ddot{\varphi }+C\left(\varphi ,\dot{\varphi }\right)\dot{\varphi }=Q$$
where $H\left(\varphi \right)\ddot{\varphi }$ is the inertial force term, $C\left(\varphi ,\dot{\varphi }\right)\dot{\varphi }$ is the nonlinear term, $\dot{H}\left(\varphi \right)\dot{\varphi }$ is the Gauche force, and $-\frac{1}{2}{\dot{\varphi }}^{T}\frac{\delta H\left(\varphi \right)}{\delta \varphi }\dot{\varphi }$ is the centrifugal force.

## 3. Planning and Control Methods

### 3.1. Event–Time Hybrid Trigger Design

Since two symmetrically distributed robotic arms need to be reserved as the operating robotic arms in the crawling process of the space dobby robot, the space robot crawls using the reaction of four robotic arms with the discontinuous environment, so it is necessary to refer to the definition of the gait of the quadruped robot to define and illustrate the motion pattern of the space robot.

$P_{1}$, $P_{2}$, $P_{3}$, and $P_{4}$ are the locations of the hand claw attachment points of the four robotic arms. Create a projection of $P_{1}$, $P_{2}$, $P_{3}$, and $P_{4}$ in the direction of velocity ***ν***, obtain the projection $l_{Pi}(i=1,2,3,4)$, and calculate the angle $\theta_{i}(i=1,2,3,4)$ to the direction of velocity. Projection $l_{Pi}(i=1,2,3,4)$ satisfies the relationship.
(9)$$l_{Pi}=\frac{\nu \cdot P_{i}}{\left|\nu \right|}$$

According to the $l_{Pi}(i=1,2,3,4)$ and $\theta_{i}(i=1,2,3,4)$ relative relationship, the correspondence between the attachment points of the claws of the four robotic arms and the motion of the substrate is determined.

Referring to the definition of the trot gait of the quadruped robot, this paper divides the four mechanical arms into two groups (Figure 3): the first and fourth mechanical arms are a group, and the other two mechanical arms are a group.

The gait-switching hybrid trigger of the space robot is referenced to the quadruped robot and improved according to the actual situation. The gait switching of the quadruped robot requires the quadrupeds to carry out high-frequency oscillating phase-supporting phase motion mode [22], and the hybrid trigger is mainly time-triggered, as in Figure 4.

Time-trigger: set the duty cycle, motion period, and other parameters to design the action in accordance with the timing. Event-trigger: the difference between the real environment and the ideal environment caused by the deviation of movement, for example, the swing leg caused by uneven ground touches the ground in advance.

For the quadruped robot with high-frequency motion, using the time-trigger mechanism as the main trigger mechanism can ensure the high response speed of the system and improve the resistance of the system to gravity and disturbance at the expense of effective energy utilization. As an auxiliary triggering mechanism, the event-triggered mechanism can cope with emergencies and disturbances in the high-dynamic system environment, and improve the stability of the system from this perspective.

However, for space robots, effective energy utilization is a very important index, and the microgravity and vacuum environment in space also reduces the influence of gravity, air resistance, and other disturbances, so the main trigger mechanism for the gait switching of space dobby robots becomes event-triggering, and gait switching is triggered by constraints such as the limit of the reachable range of the robot arm and the limit of the moving range of the end of the robot arm. At the same time, unlike the quadruped robot, which uses the time period as the basis for switching states, the space dobby robot calculates the swing phase duration based on the desired base state, as shown in Figure 5.

Event 1: the hind arm (minimum projection in velocity direction) reaches warning length. Event 2: unable to reach the designated truss point at the planning time, the robot arm grips in advance.

Compared with the hybrid trigger with time as the main trigger mechanism, the event-triggered hybrid trigger makes the motion more effective by making each robotic arm motion more adequate for the substrate movement.

### 3.2. Autonomous Planning Method

The space robot robotic arm has the characteristics of a large working space and a high degree of freedom, and the space crawling environment has the environmental characteristics of discontinuous and low gravity. In this paper, the matching problems of grasping points and alternative grasping points that take into account the task objectives and the self-collision problem of the robotic arm are realized by the dynamic artificial potential field method.

For the problem studied in this paper, the desired position of the substrate is regarded as the gravitational force source, meaning that the end of the robotic arm in the swing phase is the object under investigation, and the other three robotic arms and the substrate itself need to be used as the repulsive force source in order to reduce the mutual collision problem of the robotic arms during the motion. The complex robot arm model greatly increases the complexity of the problem; therefore, a reasonable simplified model is created, and the anti-collision model of the robot arm is equated with a repulsive point and a repulsive influence radius.

By analyzing the potential field during the current motion iteration, the point with the lowest calculated potential energy is selected as the next grasping point among the graspable points within the range of motion of the action robot arm.

The potential field model of this system uses the FIRAS function model [23]:(10)$$\left\{ \begin{array}{l} U_{att}=\frac{k_{a}{\left(p-p_{g}\right)}^{2}}{2}\\ U_{ref}=\{\begin{array}{ll} \frac{1}{2}k_{r}{\left(\frac{1}{d}-\frac{1}{d_{0}}\right)}^{2} & \;d\leq d_{0}\\ \;0 & \;d>d_{0} \end{array} \end{array}\right.$$
where $U_{att}$ is the potential energy generated by the gravitational source; $U_{ref}$ is the potential energy generated by the repulsive source; $k_{a}$ is the gravitational field coefficient; $k_{r}$ is the repulsive field coefficient; p is the system state value (this system refers to the position); $p_{g}$ is the system desired state (this system refers to the desired position); d is the distance from the object to the obstacle (this system refers to the distance between the robotic arm that will swing and the other robotic arms); and $d_{0}$ represents the radius of influence of the robotic arm collision.

### 3.3. Control Architecture

Whole-body control (WBC) is a hierarchical control approach for the redundant control problem of mobile-based robotic systems [24]. The basic idea is to decompose the motion of the robot system into task dynamic behavior and posture behavior according to the task orientation, i.e., to solve the redundant control problem of the mobile-based robot by decomposing the task and to design the joint control of interrelated and influential multi-level controllers to realize the task [25]. The schematic diagram is shown in Figure 6.

For the object studied in this paper, the hierarchical control of the whole-body controller is mainly reflected in the motion planning of the substrate of the space dobby robot and the motion planning of the gripping point of the discontinuous terrain environment (truss) of the robot arm, while the substrate motion of the space robot will be powered by the support phase robot arm, so for the support phase robot arm, the discontinuous terrain environment again puts forward motion constraints on the gripping point of the robot arm to compensate for the task. The kinetic behavior level is compensated.

The object whole-body controller design and overall control flow studied in this paper are shown in Figure 7. The whole control flow is divided into steps, including:(1)Task timing decomposition: The decomposition of task-oriented requirements and information about the desired state of the substrate during the design process.(2)Deviation calculation: Calculate the deviation of the substrate state according to the desired substrate state and the actual substrate state.(3)Dynamic selection of grasping points by potential field method: Based on the trot gait, the appropriate grasping points are calculated in real time based on the artificial potential field method.(4)Motion state decomposition: The decomposition of the state deviation of the substrate to the root of each robot arm and the calculation of the drive to be provided at the end of the robot arm according to the current support swing state of the arm.(5)Coordinate system conversion: It is necessary to unify the coordinate systems of the drives in the process. The drives in this paper are described under the ontological coordinate system, and the planning of the gripping points is described under the inertial system, which needs to be converted according to the actual state information of the substrate.(6)Cartesian space to joint space mapping: For the end position velocity of the robot arm already planned in the previous paper, inverse kinematics is applied and joint space position velocity mapping is performed.(7)Virtual model controller design: In the joint space, the joint is considered as the equivalent model of damping and spring, and the control drive is performed by calculating the equivalent effectiveness of the virtual model.(8)State feedback: The dynamic simulation model provides state feedback for the controller.

In the whole simulation control process, (1)–(4) form the whole-body controller part and (5)–(8) form the virtual model control part. The two parts are combined together to complete the control requirements.

## 4. Results

### 4.1. Description of Simulation Conditions

The equivalent parameters of the planning motion effectiveness of the manipulator are shown in Table 2. The truss parameters are shown in Table 3. The parameters of the artificial potential field are shown in Table 4.

### 4.2. Autonomous Planning Simulation Results

The planning results are shown in Figure 8. In this case, the closed quadrilateral represents the area surrounded by the four grasping points, and the blue squares in the figure represent the projection of the center of mass position. The state after each movement will be added and modified on the basis of the previous figure.

In the process of selecting the grasping points by the robot arm, the other robot arm grasping points are used as obstacles to generate the repulsive potential field, and to reduce the appearance of the phenomenon of four arms gathered in the center of mass, a virtual obstacle is also equivalently designed at the center-of-mass position for the correction of the grasping point position.

Figure 9 represents the potential field of the space dobby robot during three crawls, which gradually decreases from the yellow to the blue potential field, where the closed range in blue is the step limit of the single advance of the robot arm, and the closed range in green is the length limit of the robot arm.

The space robot takes the lowest point of the potential field within the selected constraint range as the basis for autonomous crawling planning and decomposes the task-oriented autonomy of each robotic arm to realize the autonomous planning of the space robot.

### 4.3. Virtual Prototype Simulation

Figure 10 shows the simulation process of the space robot crawling with a trot gait. Four motion state transitions are performed, considering that only two robotic arms cannot provide enough driving force when driving, and reducing the planned speed can make the motion smoother. Trusses are used as typical discontinuous terrain for space robot-dependent crawling.

Referring to Figure 11, Figure 12 and Figure 13, the effect of the base motion is analyzed. From the three figures, it can be seen that the three-dimensional displacement shows large fluctuations at 14 s, 23 s, 38 s, and 55 s times, and against the simulation playback, at the above times, the space robot performs gait switching.

The process motion error statistics are shown in Table 5. In the time period from 0 s to 23 s, the space robot position tracking effect is very good, and the displacement deviation in all three directions of XYZ is less than 0.02 m.

In the time period from 23 s to 38 s, the displacement deviation in the X-direction gradually increases to 0.08 m, the displacement deviation in the Y-direction gradually increases to 0.04 m, and the displacement deviation in the Z-direction grows from 0.01 m to 0.03 m, referring to Figure 14. This is because the support phase mechanical arm is too close to the center of mass, so the force arm becomes shorter and the driving capacity decreases.

From 38 s to 55 s, the base displacement error is smooth. The error in the X-direction is 0.02 m, the error in the Y-direction is 0.02 m, and the error in the Z-direction increases to 0.06 m at this stage. From 55 s to 60 s, the mechanical arm does not have switching behavior, the position deviation in the X-direction increases linearly, and the maximum error is 0.04 m.

From 55 s to 60 s, no switching behavior of the robot arm occurs. The position deviation in the X-direction increases linearly with the maximum error of 0.04 m, the position deviation in the Y-direction increases linearly with the maximum error of 0.02 m, and the error in the Z-direction decreases to 0.03 m.

## 5. Conclusions

In this paper, the autonomous planning problem under discontinuous terrain in the study of the space-dependent crawling scenario of the space dobby robot is addressed in the context of space missions, and the autonomous planning of space robot motion is realized by adopting the dynamic setting of artificial potential field methods to select the grasping point of the robot arm considering the motion effectiveness and self-collision of the robot arm. The simulation is verified in the dynamics software MBDyn by adopting the combination of a whole-body controller and virtual model control.

## Figures and Tables

**Figure 1 sensors-23-03334-f001:**
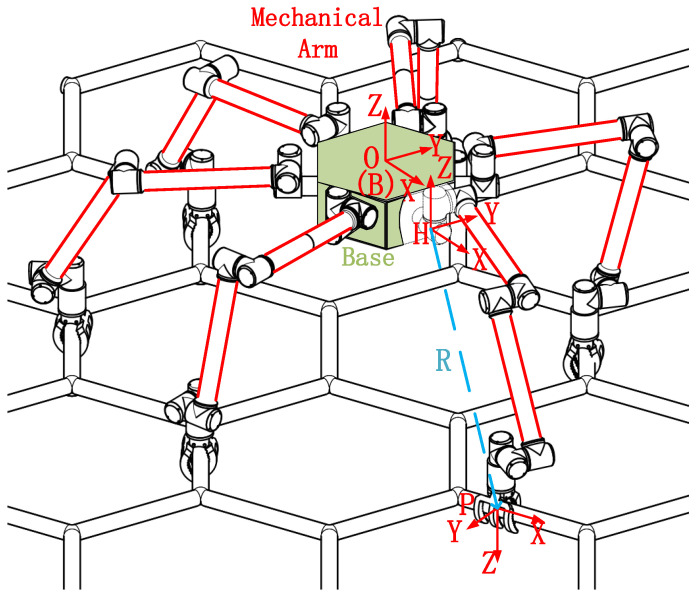
Mechanism diagram of a space dobby robot.

**Figure 2 sensors-23-03334-f002:**
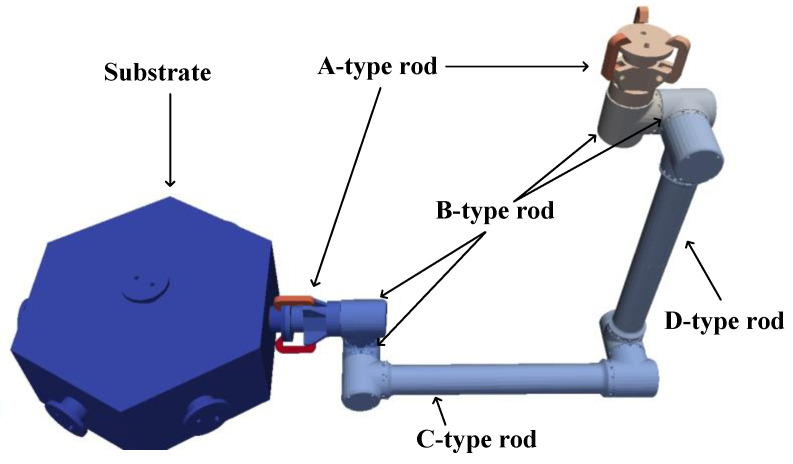
Mechanic arm configuration mechanism diagram.

**Figure 3 sensors-23-03334-f003:**
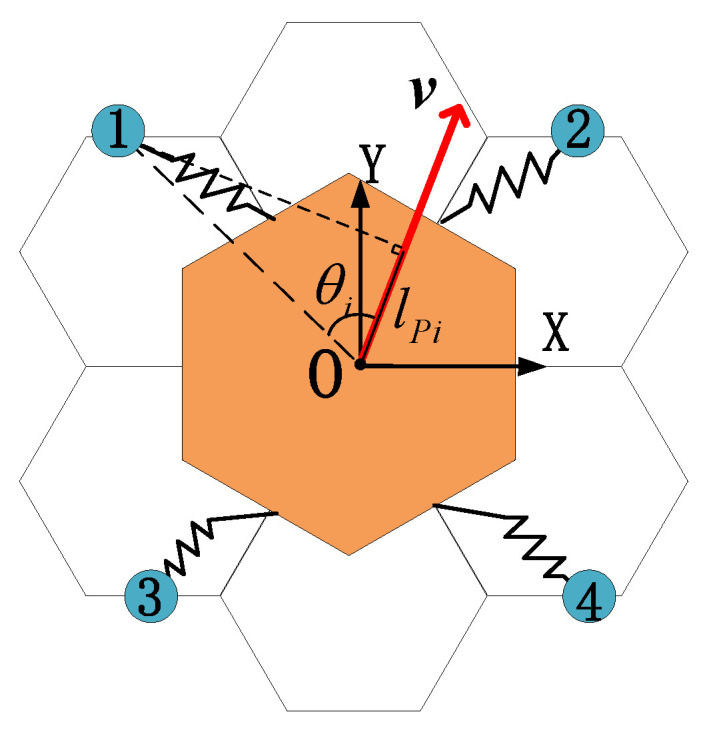
Schematic diagram of space robot robotic arm numbering. The numbers are manually calibrated to facilitate the description of relative positional relationships.

**Figure 4 sensors-23-03334-f004:**
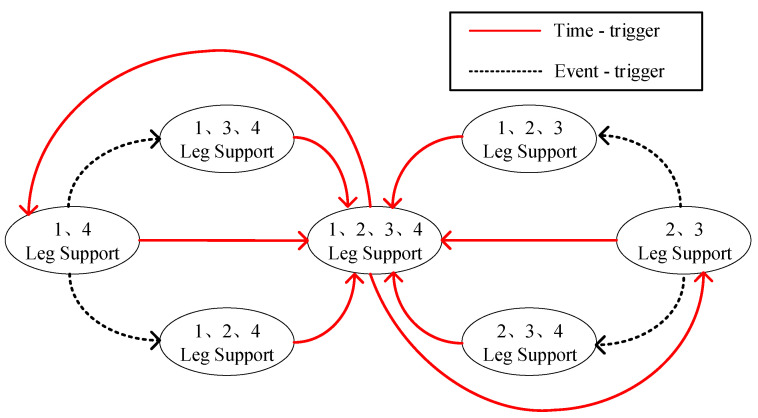
Time-triggered hybrid trigger. The numbers are manually calibrated to facilitate the description of relative positional relationships, Details are in Figure 3.

**Figure 5 sensors-23-03334-f005:**
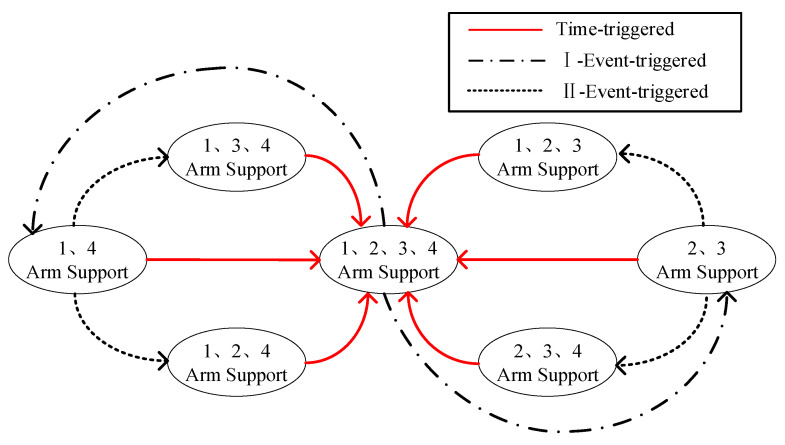
Event-triggered hybrid triggers. The numbers are manually calibrated to facilitate the description of relative positional relationships, Details are in Figure 3.

**Figure 6 sensors-23-03334-f006:**
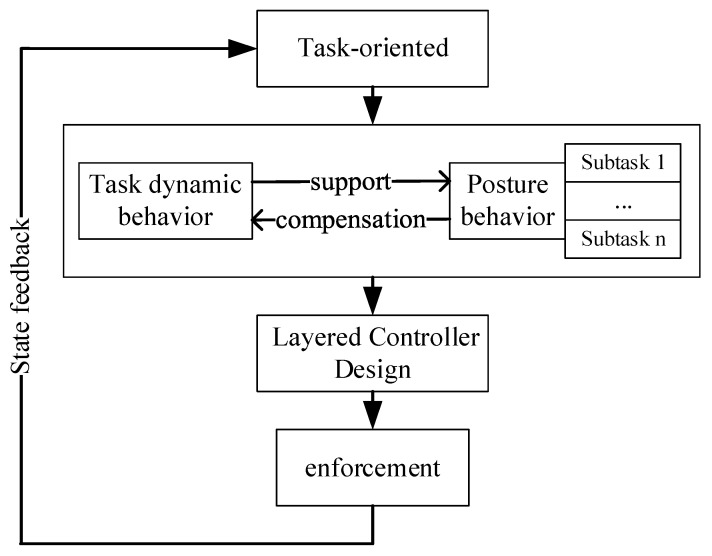
Schematic diagram of the whole-body controller.

**Figure 7 sensors-23-03334-f007:**
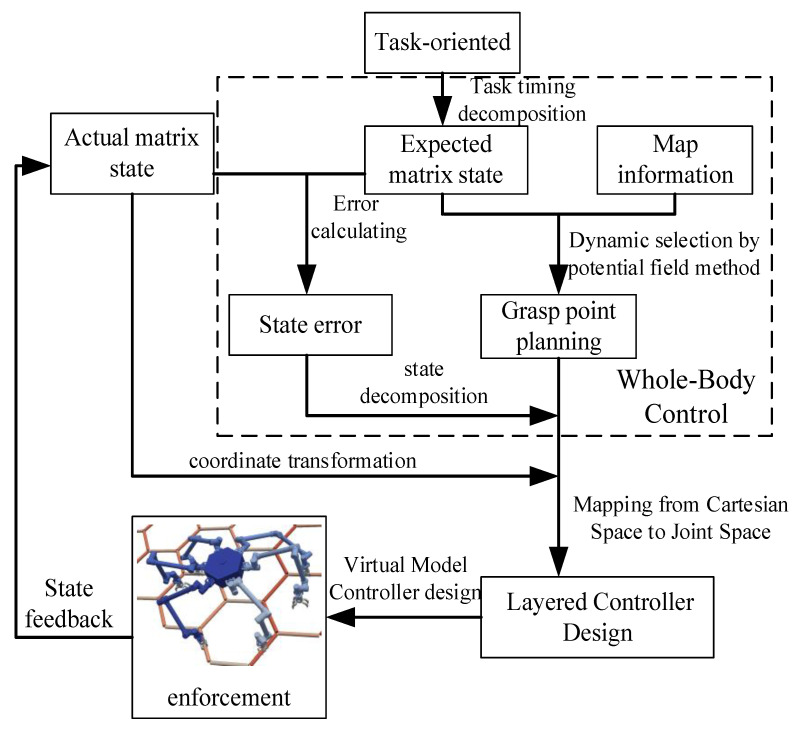
Whole-body controller design and overall control flow.

**Figure 8 sensors-23-03334-f008:**
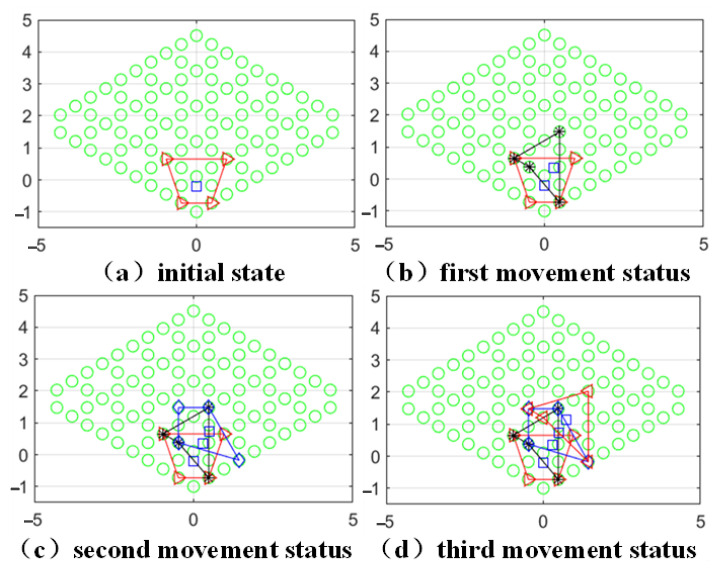
Trot gait planning motion state.

**Figure 9 sensors-23-03334-f009:**
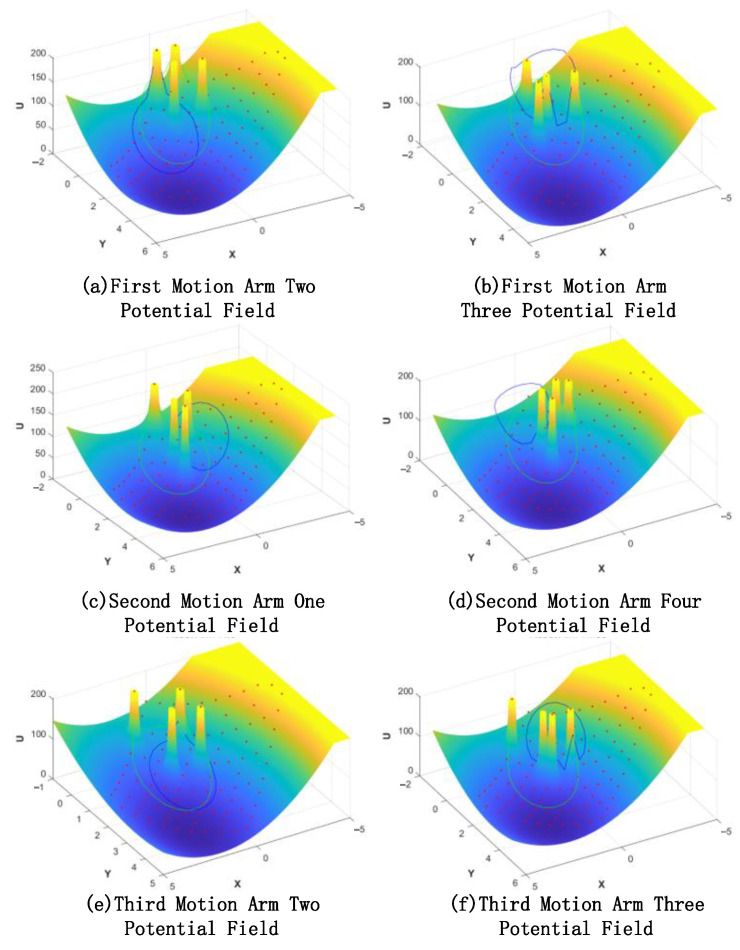
Dynamic potential field diagram with constraints.

**Figure 10 sensors-23-03334-f010:**
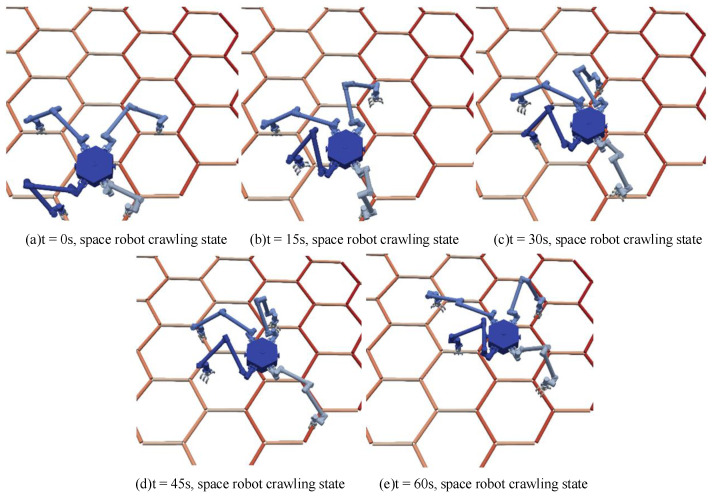
Space robot crawling process.

**Figure 11 sensors-23-03334-f011:**
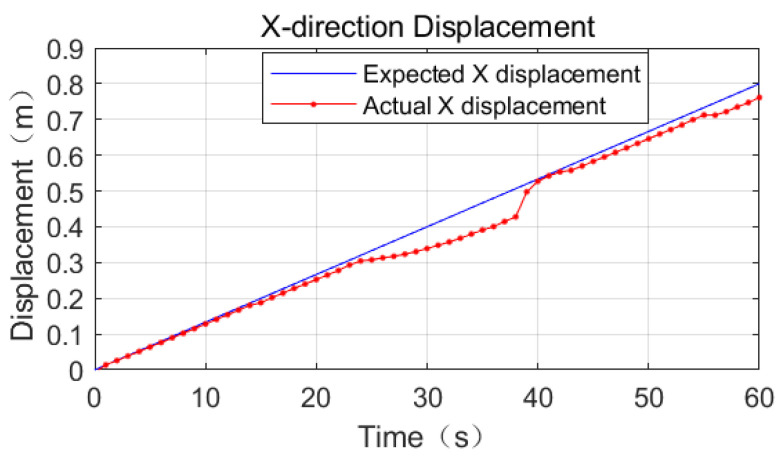
Comparison of expected and actual displacement in the X-direction of base truss crawling.

**Figure 12 sensors-23-03334-f012:**
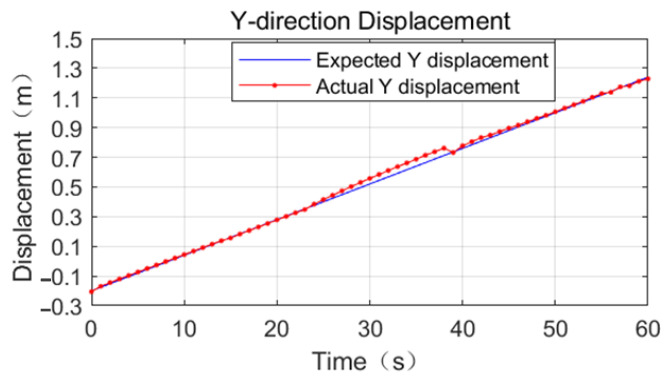
Comparison of expected and actual displacement in the Y-direction of base truss crawling.

**Figure 13 sensors-23-03334-f013:**
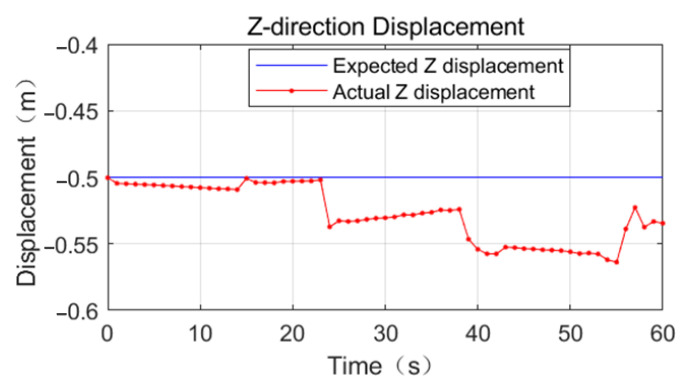
Comparison of expected and actual displacement in the Z-direction of base truss crawling.

**Figure 14 sensors-23-03334-f014:**
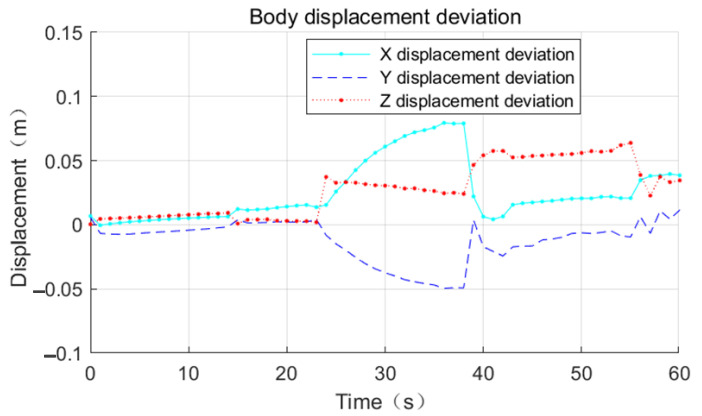
Deviation of base joist crawling displacement.

**Table 1 sensors-23-03334-t001:** Space dobby robot parameters.

Name	Parameter
Substrate	Quality: 44.277 kg Side length: 300 mm Height: 0.2 m Principal moments: [0.899,0.899,1.512] kg m^2^
A-type rod	Quality: 0.4839 kg Length: 89.3 mm Principal moments: [3.865,3.865,4.300] kg m^2^
B-type rod	Quality: 0.4839 kg Length: 99.3 mm Principal moments: [5.333,5.121,5.274] e-4 kg m^2^
C-type rod	Quality: 1.0827 kg Length: 271 mm Principal moments: [1.266,3.614,3.583] e-4 kg m^2^
D-type rod	Quality: 1.4054 kg Length: 291 mm Principal moments: [1.475,3.857,3.845] e-4 kg m^2^

**Table 2 sensors-23-03334-t002:** Space robot motion performance equivalents.

Name	Parameter
Equivalent maximum elongation length of manipulator	1.6 m
Initial position of substrate	[0, −0.2, −0.5] m
Target location	[1.6, 1.0, −0.5]
Maximum stride radius of manipulator	0.8 m
Safety velocity coefficient	0.6
Manipulator One initial position	[−0.954, 0.6524, −1]
Manipulator Two initial position	[0.954, 0.6524, −1]
Manipulator Three initial position	[−0.477, −0.7246, −1]
Manipulator Four initial position	[0.477, −0.7246, −1]

**Table 3 sensors-23-03334-t003:** Truss model parameters.

Name	Parameter
Rod number	94
Rod length	0.6 m
Rod radius	0.03 m

**Table 4 sensors-23-03334-t004:** Virtual potential field parameters.

Name	Parameter
Gravitational field gain	10
Repulsion field gain	6
Repulsion field action range	1.2 m
Potential field potential energy limit range	[0, 200]

**Table 5 sensors-23-03334-t005:** Joist crawling substrate motion status table.

Motion Iterations Number	Time	X Displacement Deviation	Y Displacement Deviation	Z Displacement Deviation
1	0–14 s	0.01 m	0.01 m	0.01 m
2	14–23 s	0.02 m	0.01 m	0.01 m
3	23–38 s	0.08 m	0.05 m	0.04 m
4	38–55 s	0.02 m	0.02 m	0.06 m
5	55–60 s	0.04 m	0.02 m	0.03 m

## Data Availability

Not applicable.

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
