# Peer review of "Autonomous Planning of Discontinuous Terrain-Dependent Crawling for Space Dobby Robots"

_sensors, 2023, doi:10.3390/s23063334_

Round 1

Reviewer 1 Report

This paper presents an autonomous planning method for space dobby robots based on dynamic potential fields. The method is triggered by hybrid event-time that combines the working characteristics of space dobby robots and improving the gait timing trigger. Effectiveness of the method verified by the computer simulation results.

The presenting topic is very interesting. The Section 1 gives a clear description about motivation, related works, and contribution of the paper. Unfortunately, some parts of the paper are not clear and difficult to be followed as given in the following comments:

1) Writing style needs to be improved.

2)  The Section 2 is not quite clear in describing the robot model. Give a brief description about how the robot works.

3)  The Figure 1 needs to use different colour to distinguish the robot and the base.

4)    What are the A-,B-,C-,D- types rod, are they just names or have specific meaning?

5)    How the space dobby robot can be regarded as a single rigid body with 6 DOF? Considering it as 6 DOF, the robot is then similar to others ordinary robot in 3D space. If we check the number of actuators, the robot should have more than 6 DOF. Simulation result in Figure 10 shows that the robot model has more than 6 DOF.

6)   Please revise the Section 2 first and then we can go further to the next sections.

Author Response

Thank you very much for your suggestion. I have revised it according to your suggestion.

Reviewer 2 Report

This article is quite interesting because, after the authors carry out a detailed bibliographical review of the last 11 years, they conclude that: “the autonomous planning of crawling motion of space dobby robots in discontinuous environments needs to be investigated”. Based on this, the authors write this article. All 25 references are cited in the article and appropriate analytical development is carried out through the dynamic potential field method, and whole-body controller design. Numerical simulation results are presented using MBDyn software.

Thus, I am in favor of publishing this article, but I suggest minor improvements, as described below:

1. (page 4) Where it reads “2.1.1 Kinematic model”, change it to “2.2.2 Kinematic model”.

2. (page 5) In the first paragraph when the meaning of the variables in eq. (1) is explained, the subscript of the variable R is “eb” and not “ib”. The authors confused this with the terms of eq. 2. Fix it!

3. (pg. 5) Where it reads “2.1.2 Kinetic modeling...” replace it with “2.2.2 Kinetic modeling...”.

4. (p. 6) Immediately before equation (9) there is a missing line in the text. Merge it with the previous paragraph.

5. (pg. 7) Avoid using paragraphs that are neither too long nor too small in the text of the article. Therefore, merge paragraph 2 with paragraph 3 and merge paragraph 6 with paragraph 5, as they are small paragraphs.

6. (pg. 9) Change “3.3 control architecture” to “3.3 Control architecture”.

7. (p. 9) In the paragraph “(1) Task timing…” after the colon, start the text in capital letters.

8. (pg. 10) Figure 7 illustrates very well the method proposed by the authors.

9. (p. 14) Place Table 5 fully on p. 14 or fully on p. 15. Do not leave it as it is (part on one page and part on another page).

10. (pg. 14) The single paragraph on this page is too long. Review the punctuation of this paragraph. Use a full stop at most every 2.5 lines. You wrote a 14-line paragraph without using any full stop.              

11. It would be interesting to give a last review in English.

Yours sincerely,

The Reviewer.

Author Response

(The authors gave the same response as above.)

Reviewer 3 Report

In this paper, the authors have worked on the autonomous planning of discontinuous terrain dependent crawling for space dobby robots. The proposed space dobby robot that operates in a discontinuous environment (i.e., research gap) is supported by a clear design and development process followed by reasonable simulation results. The following suggestions are given for the possible improvement of the proposed methodology. 

1. It is unclear how the proposed design will impact power consumption compared to the conventional approaches. 

2. It is unidentified how the operational cost of the proposed design will impact the chip area and performance in the sense of the consumption of the resources. 

3. There are a few typos and unclear sentences that need corrections for example - lines 151, 154-155, 178-180 and etc.

Author Response

(The authors gave the same response as above.)
